# UNCERTAINTY-AWARE DECODING WITH MINIMUM BAYES RISK

**Nico Daheim**[1]**, Clara Meister**[2]**, Thomas Möllenhoff**[3]**, Iryna Gurevych**[1]
[1]Ubiquitous Knowledge Processing Lab (UKP Lab)
  Department of Computer Science and Hessian Center for AI (hessian.AI)
  Technical University of Darmstadt    [2]ETH Zurich
[3]RIKEN Center for Advanced Intelligence Project, Tokyo, Japan
  `www.ukp.tu-darmstadt.de`

## ABSTRACT

Despite their outstanding performance in the majority of scenarios, contemporary language models still occasionally generate undesirable outputs, for example, hallucinated text. While such behaviors have previously been linked to uncertainty, there is a notable lack of methods that actively consider uncertainty during text generation. In this work, we show how Minimum Bayes Risk (MBR) decoding, which selects model generations according to an expected risk, can be generalized into a principled uncertainty-aware decoding method. In short, we account for model uncertainty during decoding by incorporating a posterior over model parameters into MBR's computation of expected risk. We show that this modified expected risk is useful for both choosing outputs and deciding when to abstain from generation and can provide improvements without incurring overhead. We benchmark different methods for learning posteriors and show that performance improves with prediction diversity. We release our code publicly.[1]

## 1 INTRODUCTION

Today's language models can generate fluent and coherent text. While they perform well in many scenarios, there are still instances where they fail and, for example, hallucinate factually incorrect outputs or generate harmful language (Ye et al., 2023; Bhandari & Brennan, 2023; Li et al., 2024). Previous works have shown that these behaviors are often related to out-of-distribution inputs (Ren et al., 2023) and (epistemic) uncertainty (Xiao & Wang, 2021; van der Poel et al., 2022; Fadeeva et al., 2024) which are both connected to uncertainty about the parameters of the model. Yet there is still a lack of methods that adjust for this type of uncertainty during decoding in language generation.

Minimum Bayes Risk (MBR) decoding was originally proposed for statistical machine translation (Kumar & Byrne, 2002), motivated by similar model shortcomings. The idea of MBR is to make use of the entire distribution when choosing an output, because, while the model distribution might be a good overall representation of the target distribution (Smith, 2011), individual samples might not be adequate. More recent works have shown that such problems persist with modern models (Stahlberg & Byrne, 2019; Cohen & Beck, 2019; Eikema & Aziz, 2020), precipitating the resurgence of MBR (Freitag et al., 2022). In this work, we show how a small adjustment to MBR decoding can enhance it beyond this scope and turn it into an uncertainty-aware decoding method.

In short, we modify MBR's definition of expected risk by incorporating an additional expectation over a posterior distribution over model parameters. This adjustment enables us to account for uncertainty in parameter estimates when judging the quality of different hypotheses from a model. We present different estimators for this expected risk which use multiple models from the (approximate) posterior to generate outputs.[2] Two of these estimators combine outputs at the sequence-level, i.e. full strings generated by a model, which is useful for combining the outputs of black-box LLMs for which one does not have access to output probabilities. Another estimator combines token-level distributions.

---

[1]`https://github.com/UKPLab/iclr2025-mbr-uncertainty`
[2]This approach has previously been shown to improve downstream performance in classification tasks (Blundell et al., 2015; Lakshminarayanan et al., 2017; Maddox et al., 2019; Shen et al., 2024).

Overall, we find strong evidence that accounting for weight uncertainty can improve decoding and reduce hallucinations when finetuning and pretraining from scratch, even without computational overhead. We find that improvements trend with the expressiveness of the posterior. Likely related to this, the performance of uncertainty-aware MBR is highly correlated with the prediction diversity across the combined models. We also find that weight uncertainty provides a useful signal for selective prediction, where we observe that the uncertainty-aware expected risk can be used to decide when to predict or abstain from generation. Furthermore, we show that performance scales: it improves with more models and larger hypothesis set sizes. Finally, we show the effectiveness of this framework when used to ensemble outputs from black-box LLMs.

## 2 BACKGROUND

### 2.1 PROBABILISTIC LANGUAGE GENERATION

Modern models for language generation are predominantly locally-normalized, autoregressive models of a conditional distribution over next tokens. The probability of a sequence of tokens forming a string can be determined by the product of all next-token probabilities in the sequence. Formally, given input $\mathbf{x}$ and model $p_{\boldsymbol{\theta}}$ the probability of an output sequence $\mathbf{y} = \langle y_1, y_2, \dots \rangle$ is computed as

$$p_{\boldsymbol{\theta}}(\mathbf{y} \mid \mathbf{x}) = \prod_{t=1}^{|\mathbf{y}|} p_{\boldsymbol{\theta}}(y_t \mid \mathbf{y}_{<t}, \mathbf{x}). \tag{1}$$

Here, each $y_t$ is a token from some predetermined vocabulary $\mathcal{V}$ and $\boldsymbol{\theta} \in \mathbb{R}^d$ are the parameters of the model which are also often called weights. The input $\mathbf{x}$ could be text but, for example, also images.

**Learning $p_{\boldsymbol{\theta}}$.** The parameters of the model $p_{\boldsymbol{\theta}}$ are generally learned given paired examples $\mathcal{D} = \{\mathbf{x}^{(i)}, \mathbf{y}^{(i)}\}_{i=1}^N$, a loss function and an optimization procedure. The loss function then indicates how well the model $p_{\boldsymbol{\theta}}$ captures the data-generating distribution $p(\cdot \mid \mathbf{x})$ from which we assume $\mathcal{D}$ is sampled. In most cases, language generation models are learned by minimizing an empirical risk over data examples in terms of one parameter set $\boldsymbol{\theta} \in \mathbb{R}^d$, for example, using AdamW (Loshchilov & Hutter, 2019). However, such approaches can not directly model weight uncertainty. In this work, we instead use Bayesian methods to model weight uncertainty. We describe them in §3.1 and §4.2.

**Decoding from $p_{\boldsymbol{\theta}}$.** At inference time, our goal is to generate a string from $p_{\boldsymbol{\theta}}(\cdot \mid \mathbf{x})$. The set of decision rules used in this process is often referred to as the decoding strategy. One such strategy is simply to sample tokens autoregressively until a stopping criterion, usually a fixed maximum length or a special end-of-sequence token, is met. Another strategy is to (approximately) search for the maximum probability string according to $p_{\boldsymbol{\theta}}(\cdot \mid \mathbf{x})$. Both of these approaches have proved problematic empirically (Fan et al., 2018; Holtzman et al., 2020; Eikema & Aziz, 2020; Hewitt et al., 2022), prompting the exploration of alternative strategies. The shortcomings of all of these strategies have been (at least partially) attributed to the fact that they do not consider a string's utility, which may not perfectly align with its probability. Minimum Bayes Risk decoding aims to solve this issue.

### 2.2 MINIMUM BAYES RISK DECODING

Minimum Bayes Risk decoding is derived from Bayesian Decision Theory, which states that optimal decisions are those that minimize an expected risk or, equivalently, maximize an expected utility (see DeGroot, 2005, inter alia). Given a utility function $u : \mathcal{V}^* \times \mathcal{V}^* \to \mathbb{R}_{\geq 0}$ which assigns to each pair of strings a non-negative utility, MBR aims to find the string that maximizes expected utility with respect to the target distribution. This principle is especially appealing when working with a possibly imperfect model of the target distribution, such as $p_{\boldsymbol{\theta}}$, because it allows using the full model distribution instead of relying on the adequacy of individual samples, which is argued to be the downfall of other decoding strategies (Eikema & Aziz, 2020). We thus choose the hypothesis:

$$\mathbf{y}^* = \operatorname*{arg\,max}_{\mathbf{y}' \in \mathcal{V}^*} \mathop{\mathbb{E}}_{\mathbf{y} \sim p_{\boldsymbol{\theta}}(\cdot \mid \mathbf{x})} \left[ u(\mathbf{y}, \mathbf{y}') \right] \tag{2}$$

$$= \operatorname*{arg\,max}_{\mathbf{y}' \in \mathcal{V}^*} \sum_{\mathbf{y} \in \mathcal{V}^*} p_{\boldsymbol{\theta}}(\mathbf{y} \mid \mathbf{x}) u(\mathbf{y}, \mathbf{y}'). \tag{3}$$

There are several obstacles to computing Eq. (3). Both summing over all possible strings in $\mathcal{V}^*$ to compute the expectation and searching over them to find the expectation-maximizing hypothesis are computationally infeasible.[3] Thus, approximations to Eq. (3) are used in practice.

The common approach to circumvent these obstacles is to employ an (often Monte Carlo) estimator of the expected utility and limit the search space to a subset of $\mathcal{V}^*$. Since the estimator requires a sample of strings from the distribution of interest, the same strings are often used in both the utility estimation and approximate search.[4] We refer to this collection as the hypothesis set and denote the samples used in our estimator as $\mathcal{H} = [\mathbf{y}^{(i)}]_{i=1}^N$. In the case of a Monte Carlo estimator, where all $\mathbf{y}^{(i)} \sim p_{\boldsymbol{\theta}}$, we denote this collection as $\mathcal{H}_{\boldsymbol{\theta}}$. This leads to the following approximation to Eq. (3):[5]

$$\widehat{\mathbf{y}}^* = \arg\max_{\mathbf{y}' \in \mathcal{H}_{\boldsymbol{\theta}}} \sum_{\mathbf{y} \in \mathcal{H}_{\boldsymbol{\theta}}} u(\mathbf{y}, \mathbf{y}'), \tag{4}$$

which has proven to be a useful criterion for selecting examples for knowledge distillation (Finkelstein & Freitag, 2024; Yang et al., 2024b). Most prior work on MBR has focused on making the approximation in Eq. (4) more efficient (Eikema & Aziz, 2022; Fernandes et al., 2022; Cheng & Vlachos, 2023; Vamvas & Sennrich, 2024) or on better choices for utility functions (Freitag et al., 2022) but few have considered an important underlying assumption: that $p_{\boldsymbol{\theta}}$ is a good substitute for $p$. If uncertainty over the suitable model parameters $\boldsymbol{\theta}$ (i.e. weight uncertainty) is high, e.g., when training data is limited, using a single $p_{\boldsymbol{\theta}}$ may not provide a good substitute. Bayesian modeling already provides tools to account for such uncertainty by marginalizing a distribution over possible parameters. We use this approach next to establish uncertainty-aware decoding schemes that account for weight uncertainty.

## 3   MINIMUM BAYES RISK DECODING WITH WEIGHT-UNCERTAINTY

In this section, we show how a simple change can turn MBR into an uncertainty-aware decoding method. We first introduce weight uncertainty. Then, we use it to establish an uncertainty-aware variant of MBR before presenting three practical decoding methods based on it.

### 3.1   GENERALIZING MBR WITH WEIGHT UNCERTAINTY

Often, a distribution over possible model parameters is used to model weight uncertainty based on Bayesian principles (Maddox et al., 2019; Osawa et al., 2019; Möllenhoff & Khan, 2023). According to Bayes' theorem, the probability of a parameterization $\boldsymbol{\theta}$ is $p(\boldsymbol{\theta} \mid \mathcal{D}) \propto p(\mathcal{D} \mid \boldsymbol{\theta}) \cdot p(\boldsymbol{\theta})$ where $p(\boldsymbol{\theta})$ is a prior and $\mathcal{D}$ is our data. Because calculating an exact distribution $p(\boldsymbol{\theta} \mid \mathcal{D})$ over model parameters is intractable, an approximate distribution $q(\cdot)$ is usually used. There are numerous methods one can use for obtaining $q(\cdot)$, for example, Laplace (Mackay, 1992; Daxberger et al., 2021) and variational learning (Graves, 2011; Blundell et al., 2015). We use variational learning, which we describe in §4.2.

Access to a posterior $q(\cdot)$ allows prediction by combining the outputs of multiple $p_{\boldsymbol{\theta}}$, weighted by the probability $q(\boldsymbol{\theta})$ of each parameterization $\boldsymbol{\theta}$. The resulting distribution is often referred to as the predictive posterior distribution, which we denote as $p_{\Theta}$. Empirically, this has been shown to improve calibration (Yang et al., 2024a) and uncertainty estimation (Shen et al., 2024). However, in modern language generation, it is not immediately clear how model predictions should be combined in practice. Combining predictions in probability space is difficult, because many modern LLMs are only accessible via APIs that do not return probabilities at either the token- or sequence-level. Standard Monte-Carlo-based methods avoid this issue, but they may be problematic: even for larger sample sizes, a given string would likely only be sampled once. While generations might be approximately similar, e.g., differing only in punctuation, this approach treats them as completely disparate. We now show how a natural extension of MBR provides a principled framework for combining model predictions that circumvents these issues.

---

[3]The latter problem is not unique to MBR, and faced by all maximization-based decoding strategies for autoregressive language generators. Hence, approximation algorithms are also used for these strategies.

[4]Some works have explored using different subsets for these two steps (Eikema & Aziz, 2022; Fernandes et al., 2022); we leave the exploration of the interaction of this design choice with our methods to future work.

[5]We drop the normalizing term for succinctness as it does not affect the $\arg\max$ operation.

We propose to generalize MBR by replacing the definition of $p_{\boldsymbol{\theta}}$ in Eq. (3) with the predictive posterior $p_{\Theta}$ to account for weight uncertainty. Then, the search problem is:[6]

$$\mathbf{y}^{\Theta} = \arg\max_{\mathbf{y}' \in \mathcal{V}^*} \sum_{\mathbf{y} \in \mathcal{V}^*} p_{\Theta}(\mathbf{y} \mid \mathbf{x}) u(\mathbf{y}, \mathbf{y}'). \tag{5}$$

We recover standard MBR when using the delta method to approximate $p_{\Theta}$, i.e., approximating the predictive posterior using one model parameterized by the mean of $q$ (Khan & Rue, 2023, App. C). Monte-Carlo-based approximations of MBR do not require knowledge of string probabilities but only the ability to sample from the model. Further, the utility function can be chosen as a soft matching between strings to account for similarities between samples instead of treating them as completely distinct, which addresses the aforementioned issues. We next discuss three decoding algorithms for approximately solving Eq. (5) using sequence- (§3.2) and token-level posteriors (§3.3).[7]

## 3.2 SEQUENCE-LEVEL POSTERIORS FOR UNCERTAINTY-AWARE DECODING

While autoregressive language models are trained to model a distribution over tokens, the quantity of interest is often the probability of an entire sequence $\mathbf{y} \in \mathcal{V}^*$. Therefore, it is natural to model a predictive posterior on a sequence-level by using an expectation over sequence probabilities:

$$p_{\Theta}^{(\text{seq})}(\mathbf{y} \mid \mathbf{x}) := \mathop{\mathbb{E}}_{\boldsymbol{\theta} \sim q} \left[ p_{\boldsymbol{\theta}}(\mathbf{y} \mid \mathbf{x}) \right] = \mathop{\mathbb{E}}_{\boldsymbol{\theta} \sim q} \left[ \prod_{t=1}^{|\mathbf{y}|} p_{\boldsymbol{\theta}}(y_t \mid \mathbf{y}_{<t}, \mathbf{x}) \right] \tag{6}$$

Using a sequence-level posterior to replace the model distribution in Eq. (3) admits two practical methods for (soft) model averaging, because when $u$ is bounded or non-negative[8] the order of the two expectations in Eq. (5) can be switched due to Fubini's theorem (DeGroot, 2005, Sec. 8.9; Robert, 2007, Sec. 2.3). Note that the second expectation (over models) is implicit in the definition of $p_{\Theta}^{(\text{seq})}$. This suggests using the equivalence:

$$\mathbf{y}^{\Theta} = \arg\max_{\mathbf{y}' \in \mathcal{V}^*} \sum_{\mathbf{y} \in \mathcal{V}^*} \mathop{\mathbb{E}}_{\boldsymbol{\theta} \sim q} \left[ p_{\boldsymbol{\theta}}(\mathbf{y} \mid \mathbf{x}) \right] u(\mathbf{y}, \mathbf{y}') \tag{7}$$

$$= \arg\max_{\mathbf{y}' \in \mathcal{V}^*} \mathop{\mathbb{E}}_{\boldsymbol{\theta} \sim q} \left[ \sum_{\mathbf{y} \in \mathcal{V}^*} p_{\boldsymbol{\theta}}(\mathbf{y} \mid \mathbf{x}) u(\mathbf{y}, \mathbf{y}') \right]. \tag{8}$$

Monte-Carlo estimators of Eq. (7) and Eq. (8) then correspond to either sampling a collection of generations from a set of models $\mathcal{M} = \{\boldsymbol{\theta}^{(i)} \sim q(\boldsymbol{\theta})\}_{i=1}^{M}$ sampled i.i.d from $q$ and using Eq. (4) or using Eq. (4) independently for each model before summing the utilities of each output across models. Formally, let $\mathcal{H}_{\boldsymbol{\theta}}$ denote the hypothesis set for each model $\boldsymbol{\theta}$ and $\mathcal{H}_{\mathcal{M}} = \biguplus_{\boldsymbol{\theta} \in \mathcal{M}} \mathcal{H}_{\boldsymbol{\theta}}$ the collection of all generated outputs across them. Note that both $\mathcal{H}_{\mathcal{M}}$, indicated by $\biguplus$, and $\mathcal{H}_{\boldsymbol{\theta}}$ preserve sample counts to ensure an unbiased estimator. Then, the approximate solutions become:[5]

$$\widehat{\mathbf{y}}^{\Theta} = \arg\max_{\mathbf{y}' \in \mathcal{H}_{\mathcal{M}}} \sum_{\mathbf{y} \in \mathcal{H}_{\mathcal{M}}} u(\mathbf{y}, \mathbf{y}') \tag{9} \qquad \widehat{\mathbf{y}}^{\Theta} = \arg\max_{\mathbf{y}' \in \mathcal{H}_{\mathcal{M}}} \sum_{\boldsymbol{\theta} \in \mathcal{M}} \sum_{\mathbf{y} \in \mathcal{H}_{\boldsymbol{\theta}}} u(\mathbf{y}, \mathbf{y}'). \tag{10}$$

This is convenient because it allows us to ensemble any set of LLMs given just the ability to sample from them and can easily be parallelized. No access to probabilities is required. For Eq. (10), even utility computation can be parallelized across models. There are trade-offs between both estimators, especially in terms of computational complexity, which we discuss next.

**Computational Costs.** Eq. (9) requires $(|\mathcal{M}| \cdot |\mathcal{H}_{\boldsymbol{\theta}}|)^2$ evaluations of $u$. This might be impractical for large sizes of $\mathcal{H}_{\boldsymbol{\theta}}$ but, intuitively, the larger amount of comparisons might be helpful for MBR. Eq. (10) requires only $|\mathcal{M}| \cdot |\mathcal{H}_{\boldsymbol{\theta}}|^2$ utility computations. This is faster and, especially for more costly utility functions that e.g. use LLMs as judges (Wu et al., 2025), can enable larger hypothesis set sizes.

---

[6]Here, $\Theta$ denotes all possible parameterizations $\boldsymbol{\theta}$ of the model and is used to indicate a predictive posterior.

[7]We refer to Malinin & Gales (2021, Sec. 3, App.A) for further discussion of token- and sequence-level posteriors for uncertainty estimation for autoregressive models.

[8]Many commonly used utility functions for MBR are bounded and non-negative. For example, BLEU (Papineni et al., 2002) and BERTScore (Zhang et al., 2020) return scores from 0 to 100 or 0 to 1, respectively.

**Discussion.** When preserving sample counts, Eq. (9) provides an unbiased estimate of Eq. (7). Intuitively, this is advantageous because highly probable sequences can contribute more to the decision. This differentiates ours from prior work, such as Kobayashi (2018, Alg. 1), who rather use a set union. Recent work (Farinhas et al., 2023) also uses Eq. (9) but does not explore the connection to weight uncertainty. Our methods draw parallels between MBR, which aims to minimize expected risk, and PAC-Bayes bounds (Alquier, 2024), which study the expected risk of predictive posteriors. Finally, it also helps to understand early system aggregation methods that use similar decision rules as shown here, e.g., by optimizing scalar model weights (González-Rubio et al., 2011, Eq. 8).

### 3.3 TOKEN-LEVEL POSTERIORS FOR UNCERTAINTY-AWARE DECODING

While our definition of uncertainty-aware MBR uses sequence-level Bayesian modeling, language models generally define distributions over tokens. It is thus natural to consider a predictive posterior defined over token-level distributions, i.e., by averaging token-level probabilities at each time-step:

$$p_\Theta^{(\text{tok})}(\mathbf{y} \mid \mathbf{x}) \coloneqq \prod_{t=1}^{T} \mathbb{E}_{\boldsymbol{\theta} \sim q} \left[ p_{\boldsymbol{\theta}}(y_t \mid \mathbf{y}_{<t}, \mathbf{x}) \right]. \tag{11}$$

Note, though, that for a given sequence, this approach will in general not assign the same probabilities as sequence-level ensembling.[9] Consequently, the decisions when using MBR might also be different. Since the expectation over models is intractable in Eq. (11), we use a Monte Carlo estimator. The estimator averages the token-level probabilities given by the models $\mathcal{M}$ during generation:

$$\widehat{p}_\Theta^{(\text{tok})}(y_t \mid \mathbf{y}_{<t}, \mathbf{x}) = \frac{1}{|\mathcal{M}|} \sum_{\boldsymbol{\theta} \in \mathcal{M}} p_{\boldsymbol{\theta}}(y_t \mid \mathbf{y}_{<t}, \mathbf{x}). \tag{12}$$

When sampling the hypotheses set $\mathcal{H}_\Theta$ from this distribution, i.e., sampling each token according to $\widehat{p}_\Theta^{(\text{tok})}$, an MBR estimator like the one in Eq. (5) can be used to incorporate weight-uncertainty:[5]

$$\widehat{\mathbf{y}}^\Theta = \operatorname*{arg\,max}_{\mathbf{y}' \in \mathcal{H}_\Theta} \sum_{\mathbf{y} \in \mathcal{H}_\Theta} u(\mathbf{y}, \mathbf{y}'). \tag{13}$$

There are several intuitive reasons why this could improve decoding. Perhaps the foremost is that probabilities obtained from model averaging are often better-calibrated than those of a single model (Yang et al., 2024a; Shen et al., 2024, inter alia). By averaging predictions, modeling weight uncertainty (over $\boldsymbol{\theta}$) can enable better estimates of predictive uncertainty (over $\mathbf{y}$). Since predictive uncertainty has been shown to correlate with hallucinations (Xiao & Wang, 2021), one hope would be that better estimates of it would downweigh e.g. hallucinated outputs.

**Computational Costs.** Token-level posteriors only require $|\mathcal{H}|^2$-many MBR comparisons when the hypothesis set size is equal to $|\mathcal{H}|$. Sequence-level combination requires $|\mathcal{M}| \cdot |\mathcal{H}|^2$-many comparisons for Eq. (10) or even $(|\mathcal{M}| \cdot |\mathcal{H}|)^2$-many comparisons for Eq. (9) if all hypothesis sets have the same size. However, fitting all models for token-level combination on one GPU can be hard and communication overhead is high when distributing them across GPUs. Further, token-level posteriors can not be used with black-box APIs that do not provide token-level probabilities.

### 3.4 SELECTIVE PREDICTION WITH BAYES RISK

For some inputs, for example, grammatically-incorrect strings, even a good model may not provide good predictions. Then, it can be wise to abstain from answering and, e.g., defer to a human expert instead. Selective prediction tackles this by abstaining for inputs (or outputs) that score highly in some criterion $s : \mathcal{V}^* \to \mathbb{R}$ that assigns a score for a given input $\mathbf{x}$. (Geifman & El-Yaniv, 2017; Ren et al., 2023; Kuhn et al., 2023). In practice, given $\alpha > 0$ and a test dataset $\mathcal{D}_{\text{test}}$, we only evaluate the model's answers for the top-$\lceil \alpha \cdot |\mathcal{D}_{\text{test}}| \rceil$ examples according to $s$. If $s$ is reliable, performance should improve as $\alpha$ decreases and we evaluate a smaller and smaller subset of outputs.

---

[9]This is because sequence-level modeling uses a an *expectation of products* approach while token-level modeling uses a *product of expectations* approach. Since expectation and product operations do not necessarily commute, these two ensemble definitions will, in general, assign different probabilities to the same sequence (Malinin & Gales, 2021).

Expected utility promises to be a good criterion: if we expect low utility, we should abstain from answering; if we expect high utility, we can place more trust in the model's answer. We compare different methods for using expected utility as the selective prediction criterion. We first consider the maximum-utility output in $\mathcal{H}_{\Theta}$ or $\mathcal{H}_{\mathcal{M}}$ for Eq. (13) and Eq. (10), i.e:[5]

$$s_{\text{tok}}^*(\mathbf{x}) = \max_{\mathbf{y}' \in \mathcal{H}_{\Theta}} \sum_{\mathbf{y} \in \mathcal{H}_{\Theta}} u(\mathbf{y}, \mathbf{y}') \qquad s_{\text{seq}}^*(\mathbf{x}) = \max_{\mathbf{y}' \in \mathcal{H}_{\mathcal{M}}} \sum_{\boldsymbol{\theta} \in \mathcal{M}} \sum_{\mathbf{y} \in \mathcal{H}_{\boldsymbol{\theta}}} u(\mathbf{y}, \mathbf{y}'). \tag{14}$$

Note that we can easily define a similar risk for Eq. (9) by replacing $\mathcal{H}_{\Theta}$ with $\mathcal{H}_{\mathcal{M}}$ in the definition of $s_{\text{tok}}^*(\mathbf{x})$. Another strategy is to use the expected utility *across* outputs for the given input. We can do this by averaging the utility of all outputs in the hypothesis set $\mathcal{H}_{\Theta}$ or $\mathcal{H}_{\mathcal{M}}$.[5]

$$\bar{s}_{\text{tok}}(\mathbf{x}) = \sum_{\mathbf{y}' \in \mathcal{H}_{\Theta}} \sum_{\mathbf{y} \in \mathcal{H}_{\Theta}} u(\mathbf{y}, \mathbf{y}') \qquad \bar{s}_{\text{seq}}(\mathbf{x}) = \sum_{\boldsymbol{\theta} \in \mathcal{M}} \sum_{\mathbf{y}' \in \mathcal{H}_{\boldsymbol{\theta}}} \sum_{\mathbf{y} \in \mathcal{H}_{\boldsymbol{\theta}}} u(\mathbf{y}, \mathbf{y}'). \tag{15}$$

## 4 EXPERIMENTS & RESULTS

Here, we demonstrate empirically that incorporating weight uncertainty can improve decoding. First, we provide brief experimental details and discuss how we learn weight uncertainty in §4.1 and §4.2. More details about our experiments are found in App. A. Then, we show results using prompted, finetuned and from-scratch-trained models in §4.3, where we explore different posteriors and model combination methods. §4.4 looks into the trade-off between performance and ensemble diversity and §4.5 Bayes risk for selective prediction. Finally, we show the scaling behavior of various methods in §4.6.

### 4.1 EXPERIMENTAL DETAILS

**Datasets.** We use WMT14 (Bojar et al., 2014), IWSLT14 (Cettolo et al., 2014), afroMT (Reid et al., 2021), IWSLT17 (Cettolo et al., 2017), WMT18 (Bojar et al., 2018), and WMT19 (Barrault et al., 2019) for machine translation, XSUM (Narayan et al., 2018) and SAMSum (Gliwa et al., 2019) for summarization, E2E-NLG (Novikova et al., 2017) for data-to-text generation, and STS-B (Cer et al., 2017) for scoring. For the latter, the model outputs a string representation of its numerical prediction and MBR corresponds to an empirical mean of the numerical predictions (Lukasik et al., 2024).

**Models.** We zero-shot prompt Llama-3 8B (Dubey et al., 2024), Mistral 7B (Jiang et al., 2023), Gemma-2 9B (Gemma Team, 2024a), and Qwen-2 7B (Yang et al., 2024). We finetune Gemma-2B-it (Gemma Team, 2024b) using LoRA (Hu et al., 2022) with ca. 0.9M trainable parameters. For training from scratch, we use the Transformer$_{\text{big}}$ architecture with ca. 261M parameters for WMT14 and Transformer$_{\text{base}}$ with 86M-126M parameters otherwise, following Vaswani et al. (2017).

**Metrics.** For machine translation, we use the SacreBLEU implementation (Post, 2018) of BLEU (Papineni et al., 2002), chrF (Popović, 2015), the quality estimator COMET$_{22}$ (Rei et al., 2022), and LaBSE (Feng et al., 2022) to evaluate hallucinations which has shown strong correlation with human judgements (Dale et al., 2023; Himmi et al., 2024). For Summarization and data-to-text generation we use ROUGE (Lin, 2004) and regression is evaluated using root mean-squared error (RMSE). We use FactCC for hallucination evaluation on XSUM (Kryscinski et al., 2020). For the utility function $u$ we use BERTScore (Zhang et al., 2020), except for IWSLT14 and afroMT, where we use BLEU.

### 4.2 LEARNING WEIGHT UNCERTAINTY.

We use the variational learning algorithm IVON (Shen et al., 2024) to estimate a posterior distribution over model weights and model weight uncertainty. We choose it, because each training run with IVON has only negligible overhead compared to AdamW (Loshchilov & Hutter, 2019) and gives comparable performance, as also shown in Tab. 1. It is also possible to use other Bayesian Deep Learning methods, such as, Laplace (Daxberger et al., 2021) or SWAG (Maddox et al., 2019) but we leave their exploration for future work. IVON learns a unimodal Gaussian posterior $q(\boldsymbol{\theta}) := \mathcal{N}(\boldsymbol{\theta} \mid \mathbf{m}, \boldsymbol{\Sigma})$ with mean $\mathbf{m}$ and (diagonal) covariance matrix $\boldsymbol{\Sigma}$. Setting model parameters equal to the mean of this distribution ($\mathbf{m}$) is similar to standard neural network training but $\boldsymbol{\Sigma}$ also provides an estimate of its stability. To be precise, for each parameter $m_i$ the variance $\Sigma_{ii}$ indicates how much this parameter can be changed without significant performance degradation which can be seen as a

| | IWSLT17 En-De | | | WMT18 Tr-En | | | XSUM | | | SAMSum | | E2E NLG | | STS-B |
|---|---|---|---|---|---|---|---|---|---|---|---|---|---|---|
| Method | BLEU | COMET | LaBSE | BLEU | COMET | LaBSE | R-1 | R-L | FactCC | R-1 | R-L | R-1 | R-L | RMSE |
| MBR (AdamW) | 19.93 | 76.62 | 83.47 | 14.75 | 78.20 | 76.02 | **33.63** | 25.67 | 27.50 | 46.47 | 36.21 | 67.88 | 44.41 | 0.330 |
| MBR@Mean | 19.73 | 76.60 | 83.51 | 15.27 | 78.44 | 77.12 | 33.04 | 25.19 | 23.56 | 46.17 | 35.98 | 68.74 | 45.16 | 0.284 |
| **Sequence-level - Eq. (9)** | | | | | | | | | | | | | | |
| Unimodal | 20.89 | 77.42 | 84.01 | 15.66 | 79.01 | 77.79 | 33.39 | **25.73** | 26.07 | 46.40 | 36.51 | 69.36 | 45.57 | 0.271 |
| Deep Ensemble | **21.24** | **77.94** | **84.20** | 15.63 | 79.01 | 77.60 | 33.37 | 25.68 | 27.40 | **46.71** | **36.87** | **69.56** | **45.77** | **0.269** |
| **Sequence-level - Eq. (10)** | | | | | | | | | | | | | | |
| Unimodal | 21.08 | 77.63 | 83.96 | 15.46 | 78.84 | 77.35 | 33.05 | 25.46 | 27.50 | 46.21 | 36.44 | 69.13 | 45.38 | 0.271 |
| Deep Ensemble | 21.20 | 77.91 | 84.04 | **15.69** | **79.10** | 77.56 | 33.10 | 25.50 | **32.86** | 46.14 | 36.48 | 69.19 | 45.31 | **0.269** |

Table 1: Sequence-level model combination to account for weight-uncertainty can improve the performance of a finetuned Gemma-2B model on various language generation and scoring tasks. Even simple posteriors that do not incur overhead during finetuning can give "for-free" improvements (unimodal). The number of total MBR comparisons is the same for all methods and each dataset. MBR@mean denotes decoding with a single model that is the mean of a variational distribution.

| | WMT14 En-De | | | | IWSLT14 De-En | | | | | |
|---|---|---|---|---|---|---|---|---|---|---|
| | Sampling | | Beam Search | | Sampling | | Beam Search | | MBR | Effective |
| Method | BLEU | COMET | BLEU | COMET | BLEU | COMET | BLEU | COMET | comparisons | beam size |
| MBR@Mean | 23.37 | 71.04 | 27.56 | 75.23 | 33.69 | 74.71 | 35.90 | 76.65 | 400 | 20 |
| | 24.30 | 72.15 | 27.53 | 75.18 | 34.53 | 75.18 | 36.07 | 76.76 | 1600 | 40 |
| **Sequence-level - Eq. (9)** | | | | | | | | | | |
| Unimodal | 24.31 | 72.09 | 27.52 | 75.16 | 34.59 | 75.15 | 35.78 | 76.55 | 1600 | 40 |
| Deep Ensemble | **24.70** | 72.39 | **28.99** | 76.02 | **36.03** | 75.79 | 38.30 | 78.01 | 1600 | 40 |
| **Sequence-level - Eq. (10)** | | | | | | | | | | |
| Unimodal | 24.21 | 72.15 | 27.56 | 75.21 | 34.65 | 75.20 | 35.99 | 76.67 | 1600 | 80 |
| Deep Ensemble | 24.67 | **72.58** | 28.29 | 75.70 | 35.42 | **75.84** | 37.42 | 77.69 | 1600 | 80 |
| **Token-level** | | | | | | | | | | |
| Unimodal | 23.44 | 71.36 | 27.75 | 75.19 | 33.62 | 74.68 | 35.94 | 76.66 | 400 | 80 |
| Deep Ensemble | 23.95 | 71.58 | 28.98 | **76.08** | 34.61 | 75.06 | **38.56** | **78.31** | 400 | 80 |

Table 2: Weight uncertainty improves decoding when training from scratch and using ancestral sampling and beam search. More complex posteriors (Deep Ensemble) provide better improvements. We use Transformer$_{\text{big}}$ on WMT14 and Transformer$_{\text{base}}$ on IWSLT17. Effective beam size = number of beams per model times number of models (we use four).

measure of uncertainty. We also use multiple models obtained from independent IVON training runs to form a Deep Ensemble (Lakshminarayanan et al., 2017) in order to study multimodal token- or sequence-level posteriors. This can be seen as constructing a mixture-of-Gaussian posterior with equal mixture component weights but incurs training overhead, since training time increases linearly with the number of mixture components. Unless otherwise stated, we use four models in total for MBR, i.e. $|\mathcal{M}| = 4$. For deep ensembles, we use the mean of each training run and for the unimodal method using IVON we use four samples from the posterior. For smaller models we train all parameters but for larger models we only train newly-inserted LoRA parameters $\boldsymbol{\theta}' \in \mathbb{R}^e$, following IVON-LoRA Cong et al. (2024). IVON-LoRA then learns a distribution $q(\boldsymbol{\theta}') := \mathcal{N}(\boldsymbol{\theta}' \mid \mathbf{m}', \boldsymbol{\Sigma}')$ while the original pretrained model parameters $\boldsymbol{\theta}$ remain fixed.

## 4.3 WEIGHT UNCERTAINTY & DECODING

**Weight uncertainty improves decoding.** Tab. 1 and Tab. 2 show results using finetuned Gemma-2B and Transformer models that were pretrained from scratch, respectively, on various language generation and scoring benchmarks. Results on two low-resource tasks from afroMT are found in App. B.1. For a fair comparison, we match the number of MBR comparisons, i.e. evaluations of the utility function $u$ for the estimator, with the single-model MBR baseline, as described in App. A.4.

We find in Tab. 1 and Tab. 2 that weight uncertainty improves performance across all benchmarks, even with matched compute budgets. That is, using just one model (and thereby neglecting weight uncertainty) performs worse than using multiple models sampled from the posterior and averaging their predictions. Improvements also tend to hold when compared to training with AdamW (Tab. 1). In particular, when using Eq. (9) with unimodal posteriors both training time and time needed for decoding are the same as for the single-model MBR baseline. We ensure that the time needed for decoding is the same by i) using only as many MBR comparisons as MBR@mean for our methods and ii) always using the same or smaller effective beam size, which is measured by the number of

| Method | IWSLT17 De-En | | | | WMT19 Cs-En | | XSUM | | MBR Comparisons | Effective Beam size |
| | 2 Models | | 3 Models | | 2 Models | | 3 Models | | | |
| | BLEU | COMET | BLEU | COMET | BLEU | COMET | R-1 | R-L | | |
|---|---|---|---|---|---|---|---|---|---|---|
| Single Model | 24.59 | 80.24 | 24.59 | 80.24 | 28.65 | 82.95 | 26.99 | 19.05 | 100 | 10 |
| Sequence-level - Eq. (9) | **26.66** | **81.60** | **29.12** | **83.06** | **30.60** | **84.12** | **28.27** | **20.22** | 400/900 | 20/30 |
| Sequence-level - Eq. (10) | 26.02 | 81.47 | 26.50 | 81.86 | 30.25 | 83.99 | 27.43 | 19.33 | 200/300 | 20/30 |

Table 3: Sequence-level model combination is also useful for ensembling zero-shot prompted LLMs. Eq. (9) performs better but requires more computation.

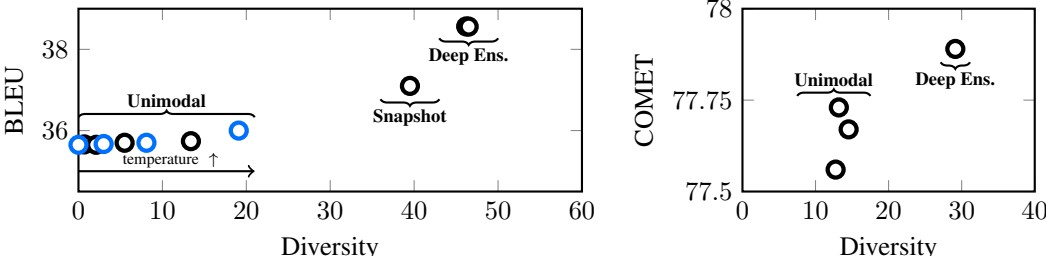

Figure 1: Our methods are more successful when the ensembled models are diverse. We compare a unimodal to mixture-based posteriors using Snapshot Ensembles and Deep Ensembles. Sampling from a unimodal posterior with higher temperature can increase diversity and improve performance (in blue). Left: token-level combination on IWSLT14 using beam search and Transformer$_{base}$. Right: sequence-level combination (Eq. (10)) on IWSLT17 using ancestral sampling and Gemma-2B.

beams per model multiplied the number of models. We validate this empirically in App. B.3. Not only do results improve when using word-overlap metrics like BLEU, but also when using quality estimation (COMET) and hallucination metrics (LaBSE). Notably, on IWSLT17 all improvements observed in COMET score when using uncertainty-aware vs. standard MBR indicate there is an estimated >85% chance that humans would distinguish the former system as better—as per Kocmi et al. (2024). Improvements also hold for the STS-B sentence similarity scoring task. The estimators of Eq. (9) and Eq. (10) perform similarly even though Eq. (9) uses a smaller hypothesis set size than Eq. (10).

**Comparison of uni- and multimodal posteriors.** Next, we compare unimodal posteriors that can be learned without overhead during training to multimodal posteriors based on Deep Ensembles. Such posteriors incur significant overhead during training, because one separate training run with different initialization and data order is required per ensemble member, but can incorporate knowledge from different loss basins—a characteristic that has proven to be beneficial (Lion et al., 2023).

When training from scratch (Tab. 2), unimodal posteriors do not consistently outperform the single model baseline when compute budgets are equivalent. In contrast, multimodal Deep Ensemble posteriors can deliver significant improvements. On the other hand, when finetuning (Tab. 1), unimodal posteriors can provide strong improvements, performing on par with Deep Ensembles. We hypothesize that this difference can be attributed to the use of LoRA for finetuning—which explores a smaller subspace of potential posterior parameters and may therefore pose a comparably easier learning problem than estimating the variance of a posterior over all parameters. Further, finetuning may not work that well for Deep Ensembles due to the models still landing in the same basin (Frankle et al., 2020). We connect our findings to prediction diversity in §4.4.

**Comparison of sequence- and token-level posteriors.** Here, we compare the use of sequence- and token-level posteriors (Eqs. (9), (10) and (13)) in MBR. Tab. 2 shows that improvements over the baseline with token-level combination are much stronger when using beam search instead of ancestral sampling to create hypothesis sets[10]. When using a mixture-based posterior, performance is improved in both settings. Sequence-level combination, on the other hand, provides similar improvements for

---

[10]Beam search provides a biased estimate and is similar to sampling from a low-temperature distribution.

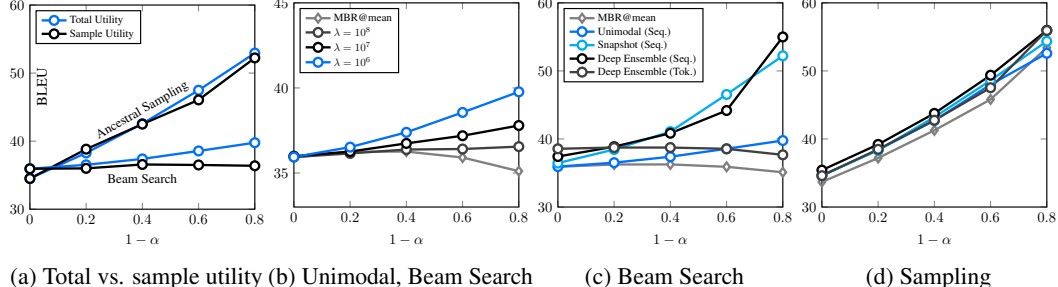

(a) Total vs. sample utility    (b) Unimodal, Beam Search    (c) Beam Search    (d) Sampling

Figure 2: Total risk and best-output-risk are useful for selective prediction. (a) Creating hypothesis sets with sampling performs better than beam search. (b) Increasing temperature when sampling from unimodal posteriors improves selective prediction. (c) When using beam search more Deep Ensembles work best. (d) For sampling, all methods work well. Results on IWSLT14 with Transformer$_{\text{base}}$.

both settings, with Eq. (9) providing similar results to token-level aggregation. Hence, the preferred method may also depend on the decoding algorithm used to create the hypothesis set.

**Ensembling zero-shot models.** Tab. 3 shows results obtained when ensembling the outputs of various zero-shot prompted LLMs on IWSLT17 De-En with a hypothesis set size of 10. We compare the estimator using an additive union of hypothesis sets (Eq. (9)) to using a soft model average (Eq. (10)) and the average single model performance. Both estimators are effective for ensembling but Eq. (9) performs best, albeit with the highest computational complexity. Details are in App. A.3.

## 4.4 CORRELATION OF QUALITY AND DIVERSITY

Next, we show that the performance of MBR with weight-uncertainty is correlated with the prediction diversity of ensembled models, potentially, due to incorporating knowledge from multiple loss basins. This is in line with prior works on ensembling which have found that diversity is important for good performance (Fort et al., 2019; Masegosa, 2020) but can form a trade-off with individual model performance (Abe et al., 2022; Wood et al., 2023).

We empirically validate this in Fig. 1, where we plot BLEU and COMET on IWSLT14 and IWSLT17 against the prediction diversity. We measure diversity as 100 minus average self-BLEU; self-BLEU scores are measured on the set of greedy decoding outputs of each ensemble member, similar to Shen et al. (2019). For finetuning, the models from the unimodal posterior are more diverse than when pretraining. The plot shows a clear correlation between both metrics. We ask two questions: 1) can diversity be promoted in unimodal pretrained posteriors to improve performance and 2) can we find a method with the same pretraining overhead as a unimodal posterior but more expressiveness?

For the first, note that the variance of the IVON posterior is $\boldsymbol{\sigma}^2 = 1/\lambda(\mathbf{h}+\delta)$, where $\mathbf{h}$ is the expected Hessian of the loss, $\delta$ is weight-decay and $\lambda$ the effective sample size which can be seen as an (inverse) temperature parameter. We decrease $\lambda$ gradually, which samples models from the posterior with higher temperature. This improves diversity and can improve performance. For the latter, we use a mixture-of-Gaussian consisting of checkpoints from one training run, denoted by "snapshot" (Huang et al., 2017). This comes at no training time increase but can improve performance by incorporating knowledge from different regions along the optimization trajectory, as shown in Fig. 1.

## 4.5 SELECTIVE PREDICTION WITH BAYES RISK

Here, we explore the use of expected Bayes risk for selective prediction on IWSLT14. We observe that both the maximum output utility and the expected output utility (i.e., average expected utility across outputs) can be used effectively for selective prediction. Our results are summarized in Fig. 2.

First, we find in Fig. 2 (a) that using the total expected utility for selective prediction performs slightly better than just using the expected utility of the chosen output. This is especially true when creating hypothesis sets with beam search, which performs much worse than ancestral sampling. Next, we

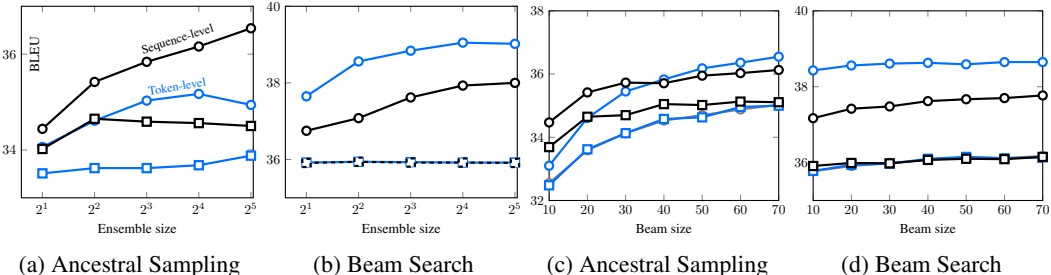

|  |  |  |  |
|---|---|---|---|
| (a) Ancestral Sampling | (b) Beam Search | (c) Ancestral Sampling | (d) Beam Search |

Figure 3: Scaling behavior on IWSLT14 with Transformer$_{base}$ in terms of ensemble (a, b) and hypothesis set size (c, d). (a, b) For a unimodal posterior (□), larger ensembles improve token-level combination using sampling but not beam search. For Deep Ensemble posteriors (○), larger ensembles generally improve performance. (c, d) Sequence-level combination (Eq. (10)) performs better for smaller beam sizes but is outperformed by token-level combination at larger ones. Scaling the hypothesis set produces stronger improvements for ancestral sampling than beam search.

again sample from the unimodal posterior with different temperatures (via decreasing $\lambda$). We find that this improves selective prediction with MBR when using beam search (Fig. 2 (b)).

Finally, we evaluate the influence of the posterior approximation. First, we find that a hypothesis set built with ancestral sampling is reliable independent of the used posterior. Even the single model baseline works well but is outperformed by using an ensemble and more expressive posteriors give bigger improvements. For beam search, the baseline completely fails and token-level combination can be unreliable. Sequence-level combination (Eq. (10)) performs much better, especially with more expressive multimodal posteriors. These results are shown in Fig. 2 (c, d).

### 4.6 SCALING BEHAVIOR

Lastly, we examine the scaling behavior of token- and sequence-level combination (Eq. (10)) with different posteriors. Results are summarized in Fig. 3. First, we show scaling the ensemble size in Fig. 3 (a) for ancestral sampling and beam search (b). Using beam search, both token- (in blue) and sequence-level (in black) combination using unimodal posteriors provide no improvements. For ancestral sampling, we find improvements with a unimodal posterior, especially at larger ensemble sizes of 32 models, but sequence-level combination of a unimodal posterior only improves until 4 models. In all other settings, scaling the ensemble size is usually beneficial.

When scaling hypothesis sets with beam search, the improvements are small, likely because the hypothesis sets lack diversity. Ancestral sampling shows a different picture and we obtain strong improvements when scaling hypothesis sets. For small hypothesis sets it is better to use sequence-level ensembling but for larger sizes token-level combination can be better.

## 5 CONCLUSION

In this work, we explore using a Minimum Bayes Risk approach to account for weight uncertainty in language model decoding. We investigate different methods which combine predictions from multiple models either during generation or afterwards. Importantly, the latter can be used to ensemble any set of LLMs. We benchmark the methods on different language generation and scoring tasks for prompted and finetuned models, as well as models trained from scratch. We show that weight uncertainty can effectively improve decoding. We evaluate the effects of using different posterior distributions. More complex distributions can sometimes provide stronger performance improvements but also simple methods can improve performance. Crucially, the improvements with simpler methods do not incur training or test-time overhead. We also connect our findings to prediction diversity, which is important for both standard MBR and when using its expected utility for selective prediction, and show that improvements scale with model and sample sizes. Overall, we find that the uncertainty-aware variant of MBR proposed in this paper leads to better and more robust language generation. Altogether, our method provides a principled approach for scaling test-time compute.

ETHICS STATEMENT

Our work uses probabilistic language models to generate language. Even when used with care, such models can produce outputs that are, among others, harmful, toxic, and hallucinated and our methods can not guarantee that such outputs are not generated. However, we aim to improve the robustness of language generation methods and, therefore, aim to alleviate these issues. Therefore, we believe there to be no direct ethical concern in our work.

ACKNOWLEDGEMENTS

This project has received funding by the German Federal Ministry of Education and Research and the Hessian Ministry of Higher Education, Research, Science and the Arts within their joint support of the National Research Center for Applied Cybersecurity ATHENE. Clara Meister was supported by a Google PhD Fellowship. This work is supported by JST CREST Grant Number JP-MJCR2112.

We thank Seyed Arshan Dalili for help with running the LLM experiments in Tab. 3.

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

# A EXPERIMENTAL DETAILS

## A.1 TRAINING FROM SCRATCH

**Datasets**  Our usage of the WMT14 English-to-German translation tasks (Bojar et al., 2014) follows the set-up from (Vaswani et al., 2017) but augments the training data by the *news-commentary-v12* data from WMT17 (Bojar et al., 2017). In total, we train on ca. 3.9M paired examples. We also use a validation set during training in order to pick checkpoints which consists of ca 39.4K examples. We use the original *newstest2014* data which consists of 3,003 examples for evaluation.

We also use the IWSLT14 German-to-English translation task (Cettolo et al., 2014) which consists of ca 160K training examples. The validation set consists of ca. 7.3K examples. The test set consists of 6,750K examples.

Furthermore, we use two language pairs from AfroMT (Reid et al., 2021), namely En-Bem (English-Bemba) which consists of 275K training, 3K validation, and 3K test examples. We do not use any monolingual data but only train from scratch on the parallel data. We use En-Run (English-Rundi) in the same way, which consists of 253K training, 3k validation, and 3k test examples.

All data usages can be reproduced by following the instructions from the Fairseq repository under `https://github.com/facebookresearch/fairseq/tree/main/examples/translation` and will be published along our code.

**Models**  All models follow the Transformer architecture from Vaswani et al. (2017) which consists of an encoder-decoder Transformer with 6 encoder and 6 decoder layers. We use the Transformer$_{\text{base}}$ architecture for IWSLT2014 and afroMT and Transformer$_{\text{big}}$ for WMT14 which has larger embedding and feed forward dimensions. The models use a vocabulary of Byte-Pair-Encoding tokens (Sennrich et al., 2016). The input and output embedding parameters of the decoder are shared. The IWSLT model has an input vocabulary size of 8848 and an output vocabulary size of 6632 for in total $39,469,056$ parameters. The en-run and en-bem models both have an input and output vocabulary size of 80000 each and a total of $126,058,496$ parameters. The WMT model has an input vocabulary size of 40480 and an output vocabulary size of $42720$ for a total of $261,431,296$ parameters.

**Training & Decoding**  We train all models from scratch using the fairseq library (Ott et al., 2019) which we extend for variational learning and a Bayesian interpretation of neural networks. Fairseq is licensed under MIT license[11] which permits our form of usage. We will release our code publicly

---

[11]`https://github.com/facebookresearch/fairseq/blob/main/LICENSE`

| Dataset | Instruction |
|---|---|
| IWSLT17 En-De | Translate from English to German: |
| WMT18 Tr-En | Translate from Turkish to English: |
| XSUM | Summarize: |
| SamSum | Summarize: |
| E2E-NLG | Convert a set of two-to-nine key-value attribute pairs in the restaurant domain to a simple English-language text: |
| STSB | How similar are these sentences from 0 to 1? |

Table 4: Simple instructions used when finetuning Gemma-2B-it.

in the future for further research in a software repository under Apache License 2.0[12]. We train all models with the IVON optimizer (Shen et al., 2024) and place a diagonal Gaussian posterior over neural networks. We use IVON with a isotropic Gaussian prior and initialize all entries of the Hessian with $0.1$. We use an effective sample size of $1 \cdot 10^{-8}$, a small weight-decay of $0.0001$, and a learning rate of $0.1$. We set $\beta_1 = 0.9$ and $\beta_2 = 0.9999$. All models are trained with a batch size of 32 or up to 1024 tokens and we use 2 MC samples from the posterior during training for afroMT and IWSLT2014. For WMT14 we just use one MC sample due to the heavier compute requirements. We clip gradients elementwise at $0.001$ and use a dropout rate of $0.2$. We train the models until performance in terms of BLEU has not improved for at least 3 epochs and then stop with the exception for WMT14, where we train only up to 20 epochs. The results for the single model baseline and unimodal posterior are averaged over four runs.

For the snapshot-like approach, we add 3 randomly-sampled distributions that were trained with at least 10 epochs to the best-performing one. For Deep Ensembles we always use four runs with different random seeds unless stated otherwise and for unimodal posteriors we sample four models from each posterior. In all experiments we sample from the posterior "as-is" and only vary the temperature by reducing the effective sample size when explicitly mentioned.

All models are trained on a single GPU which is an NVIDIA GPU with either 80GB, 40GB, 32GB or 24GB GPU memory. Training takes around 1-3 hours for the IWSLT14 and afroMT models and ¿2 days for the WMT models.

Following prior work, we use a length-penalty of $0.6$ for decoding (Vaswani et al., 2017).

### A.2 FINETUNING

**Datasets** For all datasets we use the versions from the huggingface hub (`https://huggingface.co/`). We use the En-De split of the IWSLT17 evaluation campaign (`https://huggingface.co/datasets/IWSLT/iwslt2017`) (Cettolo et al., 2017) with 206,122 training and 8079 test examples and the WMT18 Tr-En split (`https://huggingface.co/datasets/wmt/wmt18`) (Bojar et al., 2018) with 205,756 training and 3,000 test examples for machine translation. For summarization experiments, we use XSUM (`https://huggingface.co/datasets/EdinburghNLP/xsum`) (Narayan et al., 2018) and SAMSum (`https://huggingface.co/datasets/Samsung/samsum`) (Gliwa et al., 2019). XSUM has 204,045 training examples—we train only on the first $50\%$ to reduce computational load—and 11,334 test examples. SAMSum is much smaller and consists only of 14,732 train and 819 test examples. Finally, we use E2E-NLG (`https://huggingface.co/datasets/tuetschek/e2e_nlg`) (Novikova et al., 2017) with 33,524 train and 1,846 test examples for data-to-text generation, as well as STS-B (`https://huggingface.co/datasets/sentence-transformers/stsb`) (Cer et al., 2017) with 5,749 train and 1,379 test examples for sentence similarity scoring. Note that we use the version provided with the sentence transformers library (Reimers & Gurevych, 2019) which uses ratings from 0 to 1.

**Models** For finetuning results, we use the Gemma-2B-it (Gemma Team, 2024b) checkpoint, which can be found under `https://huggingface.co/google/gemma-2b-it` on the huggingface hub, with in total 2.51B parameters.

**Training & Decoding** We finetune the model using LoRA (Hu et al., 2022) with a rank $r = 8$, $\alpha = 32$ and a dropout rate of $0.1$. In total, this introduces $921,600$ new parameters that are learned with IVON and, correspondingly, the diagonal variance consists of $921,600$ further parameters that

---

[12]`https://www.apache.org/licenses/LICENSE-2.0`

| Dataset | Instruction |
|---------|-------------|
| IWSLT17 De-En | Translate the following English text to German. Make sure to only generate the translation without extra text: |
| WMT19 Cs-En | Translate the following Czech text to English. Make sure to only generate the translation without extra text: |
| XSUM | Given a BBC article, write a short summary of the article in one sentence. |

Table 5: Prompts used for zero-shot experiments.

are learned. We use the chat template provided with huggingface (Wolf et al., 2020), which we adapt to organize our experiments in line with the Apache 2.0 license it is distributed under, to organize training and decoding. As we use an instruction-tuned model, we use simple instructions for each dataset which are outlined in Tab. 4. We train the model on both the prompt and the output labels and do not only calculate gradients for the latter.

We again use IVON to learn a unimodal diagonal Gaussian posterior. We use four separate runs with different random seeds for the Deep Ensembles (which entails different data order and initialization of new parameters) and sample four models for the unimodal posterior. Results for the unimodal posterior and single model baseline are averaged over four seeds. For all experiments we use the same hyperparameter setting. We use an initial learning rate of $0.03$ which we anneal to $0$ with a cosine decay. We set $\beta_1 = 0.9$, $\beta_2 = 0.99999$, and use a small weight decay of $10^{-6}$. We again clip gradients to unit norm and element-wise with a maximum value of $0.001$. All hessian values are initialized at $0.0003$. We set the effective sample size (or inverse temperature) to $10^7$ for training but $10^9$ for decoding, because we have found this to perform better empirically, potentially due to the cold posterior effect (Wenzel et al., 2020).

For training with AdamW, we set $(\beta_1, \beta_2) = (0.9, 0.999)$ and perform a sweep over learning rates $\{1 \cdot 10^{-5}, 1 \cdot 10^{-4}, 5 \cdot 10^{-4}\}$. We again anneal the learning rates to $0$, set a small weight decay of $10^{-6}$ and rescale gradients to unit norm but do not clip them element-wise.

We train for 1 epoch for IWSLT17 and XSUM, 5 epochs for E2ENLG, 2 epochs for WMT18, and for 4 epochs on SamSUM. We always take the final checkpoints after training has ended.

### A.3 Zero-shot results

In addition to trained models, we also evaluate zero-shot prompted models. While we do not have an explicit posterior in this setting, ensembling such models can be understood as a crude approximation to sampling from the unknown Bayes posterior.

**Datasets** In addition to IWSLT17 De-En and XSUM, which are described in App. A.2, we use the Cs-En partition of WMT19 (`https://huggingface.co/datasets/wmt/wmt19`) (Barrault et al., 2019). On XSUM we only evaluate on the first 1000 examples of the test set due to computational load.

**Models** We use different models for our experiments. In particular, we use Gemma-2 9B (`https://huggingface.co/google/gemma-2-9b-it`) (Gemma Team, 2024a), Llama-3 8B (`https://huggingface.co/meta-llama/Llama-3.1-8B-Instruct`) (Dubey et al., 2024), Mistral 7B (`https://huggingface.co/mistralai/Mistral-7B-Instruct-v0.3`) (Jiang et al., 2023), and Qwen-2 7B (`https://huggingface.co/Qwen/Qwen2-7B-Instruct`) (Yang et al., 2024). We use the instruction-tuned version of each model. We select the models used for each dataset based on a manual inspection of their performance on each dataset. For example, Gemma sometimes returned czech text when asked to translate from czech to english and was therefore not included in the experiment, and Mistral tended to produce too long summaries for XSUM when compared to other models. We use the following models for each dataset: Gemma-2, Llama-3, and Mistral for IWSLT17, Gemma-2, Qwen-2, Llama-3 for XSUM, and Llama-3 and Mistral for WMT19. The prompts are shown in Tab. 5 Our prompt for XSUM is taken from (Suzgun et al., 2023).

**Decoding** We use ancestral sampling with a temperature of $1.0$ for all experiments.

| | AfroMT En-Bem | | | | AfroMT En-Run | | | | MBR | |
| | Sampling | | Beam Search | | Sampling | | Beam Search | | MBR | Effective |
| | BLEU | chrF | BLEU | chrF | BLEU | chrF | BLEU | chrF | comparisons | beam size |
|---|---|---|---|---|---|---|---|---|---|---|
| MBR (Mean) | 18.26 | 47.47 | 19.70 | 49.02 | 24.97 | 53.29 | 26.67 | 54.79 | 400 | 20 |
| | 18.63 | 47.89 | 19.70 | 49.02 | 25.58 | 53.76 | 26.67 | 54.80 | 1600 | 40 |
| **Sequence-level (Eq. (9))** | | | | | | | | | | |
| Unimodal | 18.58 | 47.84 | 19.46 | 48.88 | 25.80 | 53.86 | 26.38 | 54.65 | 1600 | 40 |
| Deep Ensemble | **19.71** | **48.77** | 21.28 | 50.35 | **26.52** | **54.56** | 28.19 | 56.02 | 1600 | 40 |
| **Sequence-level (Eq. (10))** | | | | | | | | | | |
| Unimodal | 18.43 | 47.75 | 19.62 | 48.95 | 25.34 | 53.66 | 26.58 | 54.77 | 1600 | 80 |
| Deep Ensemble | 19.48 | 48.49 | 20.69 | 49.88 | 25.86 | 54.22 | 27.40 | 55.42 | 1600 | 80 |
| **Token-level** | | | | | | | | | | |
| Unimodal | 17.90 | 47.29 | 19.60 | 48.94 | 24.86 | 53.29 | 26.57 | 54.79 | 400 | 80 |
| Deep Ensemble | 19.32 | 48.49 | **21.51** | **50.54** | 25.46 | 53.71 | **28.44** | **56.28** | 400 | 80 |

Table 6: Results on afroMT with Transformer$_{\text{base}}$ trained from scratch.

### A.4 HYPOTHESIS SET SIZES

For the finetuning experiments, we use 40 candidate hypotheses for the single model baseline and token-level combination, and 20 per model for Eq. (10) and 10 per model for Eq. (9), except for XSUM, where we use 20, 10, and 5 candidate hypotheses, respectively.

### A.5 SELECTIVE PREDICTION

For selective prediction we reuse the models and set-up from App. A.1 which were used for Tab. 2. In particular, we use the sequence-level model combination of Eq. (10) and token-level combination with both ancestral sampling and beam search. The beam size is always 40 for MBR@mean, 20 for each model used in sequence-level combination and 10 for each model used in token-level combination. All training details are the same as in App. A.1.

### A.6 SCALING EXPERIMENT

Again, we use the set-up from App. A.1 with Transformer$_{\text{base}}$ trained from scratch on IWSLT14. We scale all methods according to the same training recipe as described there but with different random seeds to train the different models.

## B ADDITIONAL RESULTS

### B.1 RESULTS ON AFROMT

Tab. 6 shows results on the En-Run and En-Bem partitions of afroMT. We find similar patterns to our results presented in Tab. 2: Deep-Ensemble-based weight uncertainty always improves performance, even with matched compute budgets, while unimodal posteriors perform similarly to a single model baseline.

### B.2 RESULTS WITH LaBSE FOR FROM-SCRATCH-TRAINED MODELS

Tab. 7 and Tab. 8 show LaBSE scores for hallucination evaluation for the same evaluation setting as in Tab. 2. Again, we find hallucinations to be reduced when weight uncertainty is accounted for.

### B.3 INFERENCE-TIME MEASUREMENTS

Tab. 9 shows the time needed for decoding in seconds as well as the obtained results for the E2ENLG experiment from Tab. 1. All results were obtained on NVIDIA GeForce RTX 3090 GPUs with 24GB memory.

| Method | Sampling | | | Beam Search | | |
|---|---|---|---|---|---|---|
| | BLEU | COMET | LaBSE | BLEU | COMET | LaBSE |
| MBR@Mean | 33.69 | 74.71 | 85.33 | 35.90 | 76.65 | 86.44 |
| **Sequence-level - Eq. (9)** | | | | | | |
| Unimodal | 34.59 | 75.15 | 85.65 | 35.78 | 76.55 | 86.42 |
| Deep Ensemble | **36.03** | 75.79 | 85.98 | 38.30 | 78.01 | 87.16 |
| **Sequence-level - Eq. (10)** | | | | | | |
| Unimodal | 34.65 | 75.20 | 85.68 | 35.99 | 76.67 | 86.45 |
| Mixture | 35.42 | **75.84** | **86.07** | 37.42 | 77.69 | 86.97 |
| **Token-level** | | | | | | |
| Unimodal | 33.62 | 74.68 | 85.39 | 35.94 | 76.66 | 86.45 |
| Mixture | 34.61 | 75.06 | 85.88 | **38.56** | **78.31** | **87.34** |

Table 7: Measuring hallucinations with LaBSE (higher is better) on IWSLT14 with Transformer$_{\text{base}}$ shows similar trends as quality estimation metrics: incorporating weight-uncertainty can reduce hallucinations, especially when a complex posterior is used. Here, we use a hypothesis set size of 20 for all methods but Eq. (9) which uses a size of 10.

| Method | Sampling | | | Beam Search | | |
|---|---|---|---|---|---|---|
| | BLEU | COMET | LaBSE | BLEU | COMET | LaBSE |
| MBR@Mean | 23.37 | 71.04 | 86.97 | 27.56 | 75.23 | 88.46 |
| **Sequence-level - Eq. (9)** | | | | | | |
| Unimodal | 24.31 | 72.09 | 87.36 | 27.52 | 75.16 | 88.42 |
| Deep Ensemble | **24.70** | 72.39 | **87.61** | **28.99** | 76.02 | 88.68 |
| **Sequence-level - Eq. (10)** | | | | | | |
| Unimodal | 24.21 | 72.15 | 87.32 | 27.56 | 75.21 | 88.44 |
| Deep Ensemble | 24.67 | **72.58** | 87.56 | 28.29 | 75.70 | 88.75 |
| **Token-level** | | | | | | |
| Unimodal | 23.44 | 71.36 | 86.84 | 27.75 | 75.19 | 88.35 |
| Deep Ensemble | 23.95 | 71.58 | 87.16 | 28.98 | **76.08** | 88.75 |

Table 8: Measuring hallucinations with LaBSE (higher is better) on WMT14 with Transformer$_{\text{large}}$ shows similar trends as quality estimation metrics: incorporating weight-uncertainty can reduce hallucinations, especially when a complex posterior is used. Here, we use a hypothesis set size of 20 for all methods but Eq. (9) which uses a size of 10.

| Method | Creation of $\mathcal{H}$ (s) | Utility Calculation (s) | Total (s) | R-1 | R-L |
|---|---|---|---|---|---|
| MBR@mean | 5824 | 402 | 6226 | 68.74 | 45.16 |
| Sequence-level Eq. (9) | 5472 | 408 | 5880 | 69.36 | 45.57 |
| Sequence-level Eq. (10) | 5881 | 418 | 6299 | 69.13 | 45.38 |

Table 9: Time (in seconds) taken for decoding for the results on E2ENLG from Tab. 1.

