# OpenReview forum: "Uncertainty-Aware Decoding with Minimum Bayes Risk"
_ICLR.cc/2025/Conference — ICLR 2025 Poster_

### Official Review · Reviewer_nT47 · 2024-10-17

**Soundness:** 3
**Presentation:** 3
**Contribution:** 3
**Rating:** 5
**Confidence:** 4

**Summary:**

This paper extends Minimum Bayes Risk (MBR) decoding to incorporate model uncertainty. During the computation of expected risk, standard MBR decoding assumes a model with fixed parameters that approximates the target distribution. This paper argues that the uncertainty in model parameters should also be considered, and proposes to compute the expected risk based on predictive posterior distribution. Specifically, the paper introduces two types of posterior distributions for uncertainty-aware MBR decoding: sequence-level and token-level posteriors. Empirical evaluation across various text generation tasks, including machine translation and text summarization, shows that the proposed method achieves consistent and strong improvements over unimodal MBR decoding.

**Strengths:**

1. This paper is well written, clearly motivated, and theoretically grounded.
2. The proposed method shows consistent and strong improvements to unimodal MBR decoding.

**Weaknesses:**

1. MBR is not a very efficient decoding algorithm. I am afraid that efforts on MBR decoding may have little chance to have practical applications.
2. The concept of token-level posterior is not well-established from a theoretical perspective. As these models represent sequence-level probability distributions, their ensemble, the predictive posterior should only be an expectation over sequence probabilities.
3. The baseline of standard MBR decoding is missing. Although there is a baseline of MBR@mean, it is not equivalent to the standard MBR decoding, where the model should be trained without weight uncertainty.

**Questions:**

Could you elaborate the difference between the two estimators in Eq.(9) and Eq.(10), and why Eq. (10) has lower complexity?

---

> ### Author Response · Authors · 2024-11-20
> **Thank you for your review!**
>
> Dear Reviewer,
> Thank you for your review! Below we address your concerns.
>
> > Q1: “MBR is not a very efficient decoding algorithm. I am afraid that efforts on MBR decoding may have little chance to have practical applications.”
>
> A1: In many applications it can still be beneficial to trade test-time compute for better performance and our method provides a well-founded approach for this. Overall, our work aligns well with current trends to scale up compute not only during training but also during testing/inference, see for example GPT-o1’s “thinking-mechanism” ((https://openai.com/index/introducing-openai-o1-preview/)). While it is true that MBR is often slower than other decoding approaches, such as greedy decoding, we note that there are various methods to improve its speed that can be combined with our work, such as the work by Cheng & Vlachos 2024.
>
> > Q2: “The concept of token-level posterior is not well-established from a theoretical perspective. As these models represent sequence-level probability distributions, their ensemble, the predictive posterior should only be an expectation over sequence probabilities.”
>
> A2: While we agree that sequence-level probabilities are in general often of greater interest (as we discuss in L175-177), we note that language models are generally estimating token-level probabilities, which are then multiplied to obtain sequence-level probabilities autoregressively. Therefore, we believe that also the token-level averaging method is theoretically sound, as is also discussed in Malinin & Gales 2021.  However, a more accurate Bayesian modeling of sequence-level probabilities may give further improvements and is an interesting direction for future work. Thank you for the suggestions!
>
> > Q3: “The baseline of standard MBR decoding is missing. Although there is a baseline of MBR@mean, it is not equivalent to the standard MBR decoding, where the model should be trained without weight uncertainty.”
>
> A3: While it is true that also the MBR@mean baseline is obtained by using a training method (IVON) that models weight uncertainty, our goal was to compare using a single model to using multiple models to show the effect of also accounting for such weight uncertainty during inference (by combining predictions of multiple models). Training a single model with AdamW will give similar results (L308-310) but is not directly comparable. We will consider adding a small additional ablation study to the final version of the paper.
>
> > Q4: “Could you elaborate the difference between the two estimators in Eq.(9) and Eq.(10), and why Eq. (10) has lower complexity?”
>
> A4: The difference is that Eq. 9 concatenates the hypothesis sets of the individual models, each of size $|\mathcal{H}_\theta |$, to a total size of $|\mathcal{M}|\cdot |\mathcal{H} \theta|$. Then, each pair out of the $|\mathcal{M}|\cdot |\mathcal{H} \theta|$-many
> hypotheses are compared for MBR, yielding $(|\mathcal{M}|\cdot |\mathcal{H} \theta|)^2$ comparisons  . Eq. 9 does these comparisons independently for each model and each $\mathcal{H} \theta$. Only afterwards there is a summation of utilities for each hypothesis and each model, and therefore the complexity is lower with $|\mathcal{M}|*|\mathcal{H} \theta|^2$ comparisons.
>
> ## References
>
> [1] Julius Cheng and Andreas Vlachos. Faster minimum Bayes risk decoding with confidence-based pruning. EMNLP, 2023.
>
> [2] Andrey Malinin and Mark Gales. Uncertainty estimation in autoregressive structured prediction. ICLR, 2021

---

> > ### Author Response · Authors · 2024-11-25
> > **Gentle Reminder.**
> >
> > Dear Reviewer, since the discussion period will be closing soon, please let us know if there are any further questions.

---

> > ### Comment · Reviewer_nT47 · 2024-11-27
> >
> > Thanks for your responses, which has addressed some of my concerns. However, I still posit that MBR decoding is not the optimal approach for trading test-time compute for better performance. Compared to other methods, MBR decoding significantly increases computational time by several orders of magnitude without delivering sufficient performance improvements.

---

> ### Author Response · Authors · 2024-12-02
> **Thank you for your response!**
>
> Dear Reviewer,
>
> Thank you for your response!
> Please note that we do not claim that MBR is always the best-performing approach in practice but we believe that MBR is useful for several reasons beyond our initial response. First, many approaches for trading test-time compute are already performing MBR [1] or can be combined with it. For example, it is possible to decode multiple numerical predictions (e.g. in math word problem solving) using chain-of-thought-based approaches and then MBR can be used to aggregate these predictions by majority voting or averaging (similar to STSB in Tab. 1). Second, if sampling multiple outputs from an LLM is too expensive during inference, MBR can still be used for distillation during both finetuning [2, 3] and pretraining [4] to improve e.g. the greedy decoding performance of LLMs.
>
> ## References
>
> [1] Amanda Bertsch, Alex Xie, Graham Neubig, and Matthew Gormley. 2023. "It’s MBR All the Way Down: Modern Generation Techniques Through the Lens of Minimum Bayes Risk." In Proceedings of the Big Picture Workshop, pages 108–122, Singapore. Association for Computational Linguistics (2023).
>
> [2] Mara Finkelstein, Subhajit Naskar, Mehdi Mirzazadeh, Apurva Shah, Markus Freitag. "Mbr and qe finetuning: Training-time distillation of the best and most expensive decoding methods." ICLR (2024).
>
> [3] Guangyu Yang, Jinghong Chen, Weizhe Lin, and Bill Byrne. 2024. "Direct Preference Optimization for Neural Machine Translation with Minimum Bayes Risk Decoding." In Proceedings of the 2024 Conference of the North American Chapter of the Association for Computational Linguistics: Human Language Technologies (Volume 2: Short Papers), pages 391–398, Mexico City, Mexico. Association for Computational Linguistics (2024).
>
> [4] Finkelstein, Mara, David Vilar, and Markus Freitag. "Introducing the NewsPaLM MBR and QE dataset: LLM-generated high-quality parallel data outperforms traditional web-crawled data." arXiv preprint arXiv:2408.06537 (2024).

---

### Official Review · Reviewer_xUiD · 2024-11-03

**Soundness:** 3
**Presentation:** 2
**Contribution:** 2
**Rating:** 5
**Confidence:** 4

**Summary:**

Minimum Bayes Risk Decoding (MBR) has a long history in machine translation and speech recognition research. As a variation of the Maximum A Posteriori decoding, it introduces a risk at the sentence level. The goal is to improve the output sentence globally. In this paper, The risk becomes utility and the goal is to find the hypothesis that maximizes the expected utility.  This paper proposes to extend this framework to include uncertainty on the parameters of the model. Therefore it requires a "double" expectation (or an expectation wrt the joint distribution of the sequence and the parameters).  Experiments are carried out on different tasks of conditional text generation: machine translation, summarization and data-to-text.

**Strengths:**

The motivation of Bayes inference is to marginalize what is uncertain. This paper proposes to take into account the uncertainty on the parameters on the top of MBR. This idea is very interesting and can be applied to both large pre-trained models, and smaller models trained from scratch only for one task.

**Weaknesses:**

The clarity is clearly the main issue with this paper; It is true that Bayesian methods are always difficult to describe, but in this paper it is sometime difficult to follow. One reason maybe that some important information are scattered in the paper.

For instance, the capital theta is not clearly defined. We can deduce it from the context but it does not help to understand. More importantly, there are many equations in the paper (this is not an issue of course) and we can get lost among their variants: the theoretical description vs  the approximated version, ...
Maybe it could help to first describe the theoretical part of the method, with the probability distributions, and then in a following part how it is proposed to approximate the intractable summations.

**Questions:**

- It could help to better describe how the q distribution on large pre-trained model is estimated.
- It looks like uncertainty improves over MBR, but can you clearly state in the table what are  the baseline systems (w/o MBR) and what are your improved ones ?
- The empirical results and the associated discussion could be improved. When you say "We find that weight uncertainty improves performance across all benchmarks, even with matched
compute budgets.", it could help to ground this claim in your tables.
- The paragraph on the complexity is a bit short. MBR decoding is costly. Maybe you solution does not increase the complexity but could you provide inference time ?
- Do you think the improvement in performance is worth the  overall computational cost ? This question stands for MBR and you method.
- When you say "Connected to this, since predictive uncertainty has been shown to correlate with hallucinations (Xiao & Wang, 2021), ...", this not the same uncertainty ! In this work they use the entropy of the word distributions not on parameters. Could you comment on that ?

---

> ### Author Response · Authors · 2024-11-20
> **Thank you for your review!**
>
> Dear Reviewer,
>
> Thank you for your review and interesting questions!
> We will use your comments to improve the writing and clarity of our work.
>
> > Q1: “For instance, the capital theta is not clearly defined. We can deduce it from the context but it does not help to understand.”
>
> A1: We have clarified the description of $\Theta$ in Footnote 4. In short, $\Theta$ refers to the space of possible model parameterizations $\theta$. We use it to make it clear that the distribution $p_\Theta$ is a predictive posterior and not based on just a single model.
>
> > Q2: “It could help to better describe how the q distribution on large pre-trained model is estimated.”
>
> A2: Thank you for the suggestion, we have improved this part of our paper in L313-317. In short, we simply run IVON together with LoRA. This means that the parameters of the pretrained model stay fixed and the distribution $q$ is only learned over the newly inserted LoRA parameters.
>
> > Q3: “When you say "We find that weight uncertainty improves performance across all benchmarks, even with matched compute budgets.", it could help to ground this claim in your tables.”
>
> A3: Thank you for the suggestion, we have added a better description of this to L354-360. In Table 1 all results use the same compute budget for inference, which we measure by the number of MBR comparisons. In Table 2 this number is explicitly shown in the column “MBR comparisons”.
>
> > Q4: “Maybe you solution does not increase the complexity but could you provide inference time ?”
>
> A4: Thank you for this question. We will consider adding a more extensive ablation of this to the final version and provide numbers here for running MBR on the full test set of E2E NLG for Table 1. These show negligible overhead for processing the MBR utility function evaluations:
>
> |  MBR@mean | Eq. 9  |  Eq. 10  |
> |---|---|---|
> |  402s | 408s  | 408s  |
>
> Creating the hypothesis sets takes the following times:
>
> |  MBR@mean | Eq. 9  |  Eq. 10  |
> |---|---|---|
> |  5824s | 5472s  | 8936s  |
>
> We use the unimodal posterior which gives the following results in terms of ROUGE-L:
>
> |  MBR@mean | Eq. 9  |  Eq. 10  |
> |---|---|---|
> |  45.16 | 45.57  |  45.38 |
>
> Note that the overhead for Eq. 10 is due to the larger effective beam size at the matched number of utility function evaluations and smaller variations may be expected due to the cluster set-up (e.g. between MBR@mean and Eq.9).
>
> > Q5: “Do you think the improvement in performance is worth the overall computational cost ? This question stands for MBR and you method.”
>
> A5: We believe it is worth it for many applications. Our method provides a theoretically well-founded way to trade-off test-time compute for better performance. This aligns well with current trends to scale up compute not only during training but also during testing/inference, see for example GPT-o1’s “thinking-mechanism” (https://openai.com/index/introducing-openai-o1-preview/).
> In many applications, such as offline translation, computations can be run on larger compute and realtime processing may not be as important.
> However, MBR might be too costly for some applications, for example, computations on edge devices, where realtime processing is important.
>
> > Q6: “When you say "Connected to this, since predictive uncertainty has been shown to correlate with hallucinations (Xiao & Wang, 2021), ...", this not the same uncertainty ! In this work they use the entropy of the word distributions not on parameters. Could you comment on that ?”
>
> A6: Thank you for the question, it is indeed true that their work aims at predictive uncertainty and ours mainly focuses on weight uncertainty, which are not the same. However, please note that the quoted lines refer to the output probabilities over token-level predictions that are obtained using an averaging of the output probabilities of multiple models. Therefore, the statement indeed refers to predictive uncertainty, similar to Xiao & Wang 2021. We have clarified this in L239-242 to avoid further confusion.

---

> > ### Author Response · Authors · 2024-11-25
> > **Gentle Reminder.**
> >
> > Dear Reviewer, since the discussion period will be closing soon, please let us know if there are any further questions.

---

### Official Review · Reviewer_TRc3 · 2024-11-04

**Soundness:** 3
**Presentation:** 3
**Contribution:** 3
**Rating:** 8
**Confidence:** 3

**Summary:**

This work proposes minimum Bayes' risk (MBR) decoding considering the uncertainty of model parameters. The MBR is a decision rule for decoding which maximizes the expected utility, not a maximum a posteriori estimate, to avoid the model errors. This work takes into account the uncertainty in model parameters so that the decision rule can reflect the uncertainty in the model parameter estimation. This work also proposes a token-level sampling but approximated by Monte Carlo in order to avoid complex computation by taking average in each step. Experiments are carried out on several natural language tasks, e.g., machine translation and summarization, and show that the proposed method achieves gains when compared with a conventional MBR method.

**Strengths:**

- This work demonstrate the use of MBR considering the uncertainty in model parameters. The idea is straightforward and the proposed approach of two decision rules is a sound method in performing either summation across samples drawn from different parameters (Eq. 9) and summing within samples from the same parameters (Eq. 10).

- The proposed approach and the experimental design is coupled with IVON optimization which take into account the variance of parameters in training. Since the variance is also estimated during training, it seems to be easy to draw samples from sampled parameters.

**Weaknesses:**

The detail of the experimental design is unclear.

- The number of samples is unclear. The line 352 stats that is is using ensemble and samples for 4, but it probably implies the number of samples from a single model parameter, and this work has no report on how many samples were drawn for model parameters, that is the size of $\mathcal{M}$. Does it mean that 1 sample was drawn for the proposed method for each parameter sample set drawn from IVON so that the proposed approach is comparable to the baseline MBR with 4 samples?

- The relation of Deep Ensemble and the proposed method is unclear. The lines 311-319 state that Deep Ensemble is performed for each training run, so that multiple IVON parameters are learned. In this case, only a single parameter set is drawn for each learned IVON model, and performed MBR decoding using the proposed method, but is it correct? Note that Deep Ensemble is performing ensemble of decoding from multiple model parameters, and thus, it is basically the same as the token-level inference in Eq. 11. When mentioning Deep Ensemble, is it performing the token-level average for each step? Or, does this work leverage Deep Ensemble to learn several parameters to compare with IVON, which employs only a single training run?

**Questions:**

Please check my comments regarding weaknesses.

- Thank you for the clarification, and I feel this work has an interesting contribution in the field.

---

> ### Author Response · Authors · 2024-11-20
> **Thank you for your review!**
>
> Dear Reviewer,
>
> We thank you for your review and address your questions below.
>
> > Q1: “The number of samples is unclear. The line 352 stats that is is using ensemble and samples for 4, but it probably implies the number of samples from a single model parameter, and this work has no report on how many samples were drawn for model parameters, that is the size of M. Does it mean that 1 sample was drawn for the proposed method for each parameter sample set drawn from IVON so that the proposed approach is comparable to the baseline MBR with 4 samples?”
>
> A1: Thank you for the question! To clarify, the original L352 indeed refers to the size of $\mathcal{M}$. We use 4 different sets of model parameters, i.e. $|\mathcal{M}|=4$, for our methods. The baseline only uses one set of model parameters. We have clarified this further and moved it to L312-317 to avoid confusion. For each of the four models in our methods we generate n output samples for the MBR hypothesis set (i.e. $|\mathcal{H}_\theta|=n$), where the number of samples varies depending on the experiment and estimator, as is described in the Appendix A.4 “Hypothesis Set Sizes”.
> For example, for Table 1 (with the exception of XSUM) we generate $n=40$ output samples for the single model, $n=20$ output samples for each of the model samples in $\mathcal{M}$ for Eq. 10 and $n=10$ output samples for each of the model samples in $\mathcal{M}$ for Eq. 9 to match the number of MBR comparisons.
>
> > Q2: “The relation of Deep Ensemble and the proposed method is unclear.”“
>
> A2: Our method in general relies on using an ensemble of models to generate multiple outputs that are combined for MBR. Using just IVON and Deep Ensembles created from IVON runs both constitute two different ways of obtaining such ensembles. The difference between the two approaches is that the posterior learned by IVON is a single Gaussian, whereas Deep Ensembles of multiple IVON runs can be seen as a mixture-of-Gaussian posterior, as we describe in L301-312.
>
> > Q3: “The lines 311-319 state that Deep Ensemble is performed for each training run, so that multiple IVON parameters are learned. In this case, only a single parameter set is drawn for each learned IVON model, and performed MBR decoding using the proposed method, but is it correct?”
>
> A3: For IVON, we draw 4 models from the approximate posterior and for Deep Ensembles we train 4 times with IVON and then use the mean of each training run for an ensemble of in total $|\mathcal{M}|=4$ models. We have clarified this in L312-317.
>
> > Q4: “When mentioning Deep Ensemble, is it performing the token-level average for each step?”
>
> A4: While we indeed also use Deep Ensembles for token-level methods (cf. Table 2), we especially use them for the sequence-level methods, for example, in Table 1 and Table 2. There, we generate one hypothesis set per model independently, and do not perform any token-level averaging at any step during generation. Rather, In the end, we combine the hypothesis sets which consist of full model generations.
>
> > Q5: “Or, does this work leverage Deep Ensemble to learn several parameters to compare with IVON, which employs only a single training run?”
>
> A5: Exactly, our main goal is to compare different forms of posteriors, as described above. Thank you for the question!

---

> > ### Author Response · Authors · 2024-11-25
> > **Gentle Reminder.**
> >
> > Dear Reviewer, since the discussion period will be closing soon, please let us know if there are any further questions.

---

> > ### Comment · Reviewer_TRc3 · 2024-11-26
> >
> > Thank you for your clarification, and the updated manuscript looks better to me.

---

> > > ### Author Response · Authors · 2024-12-01
> > > **Thank you!**
> > >
> > > Dear Reviewer,
> > > Thank you for your response and raising your score!

---

### Official Review · Reviewer_Xat4 · 2024-11-09

**Soundness:** 2
**Presentation:** 2
**Contribution:** 2
**Rating:** 3
**Confidence:** 3

**Summary:**

This paper examines Minimum Bayes Risk decoding for generated text while also ensembling together multiple models (or parameter settings).

**Strengths:**

- The paper has a fairly extensive number of experiments.

**Weaknesses:**

- I'm quite familiar with MBR, ensembling, and other related methods, and to be honest the papers seemed like it was made overly complex/mathy while still omitting details about important parts. For instance, explaining the fact that you're sampling from multiple models and doing MBR over the resulting samples (equation 10) is not a difficult concept to explain, but it was couched in lots and lots of verbose equations. On the other hand, important parts that are non-trivial, such as the IVON method introduced in section 4.1, were not explained sufficiently. I feel like the paper could benefit significantly by trying to explain things in simpler/more straightforward terms.
- There is no discussion of time or memory complexity of the methods. I expect that these methods require significant computation, so it would have been better if that could be discussed.
- It's not very clear whether the methods proposed here significantly moved forward the state-of-the-art. As far as I know, the BLEU scores on WMT14 are quite low compared to the state-of-the-art for single models, which is 35 or so.

**Questions:**

None

---

> ### Author Response · Authors · 2024-11-20
> **Thank you for your review!**
>
> Dear Reviewer,
>
> Thank you for your review and suggestions. We will use them to improve the clarity of our writing. Below we address your points directly.
>
> > Q1: “[E]xplaining the fact that you're sampling from multiple models and doing MBR over the resulting samples (equation 10) is not a difficult concept to explain, but it was couched in lots and lots of verbose equations.”
>
> A1: Thank you for the suggestion, we have rewritten L190-195 to provide a more intuitive description before formally describing the method.
>
> > Q2: “On the other hand, important parts that are non-trivial, such as the IVON method introduced in section 4.1, were not explained sufficiently.”
>
> A2: Our method does not depend on IVON in particular but only uses it as one way to learn a distribution over weights. We choose it, because it is easy to use as a drop-in replacement for AdamW with similar performance and speed (L307-309), and because the definition of variance in IVON (L450-457) allows us to easily sample model weights from the posterior distribution with varying amounts of temperature. However, any (approximate) Bayesian approach can be used, for example, a Deep Ensemble of models trained with AdamW, Laplace approximations, or SWAG. We have clarified this in L302-304, thank you for the suggestion.
>
> > Q3: “It's not very clear whether the methods proposed here significantly moved forward the state-of-the-art.”
>
> A3: Our goal is to provide an uncertainty-aware decoding method using MBR. All experiments show that this provides improvements over regular MBR (using just one model) without computational overhead during inference (L354-365) which we believe clearly demonstrates the value of our method.
>
> > Q4: “As far as I know, the BLEU scores on WMT14 are quite low compared to the state-of-the-art for single models, which is 35 or so.”
>
> A4: Our goal in the WMT14 experiment is to provide results when pretraining from scratch and not to beat LLMs that are much larger and use much more data than our Transformer-big model, which we train only on the parallel WMT14 data. We only train for up to 20 epochs due to limited compute resources (L966-970) but still closely match the results reported in Vaswani et al. 2017. We agree that scaling our work to even larger models is an interesting direction for future work but would expect similar improvements for larger state-of-the-art models, as is hinted at by the results with larger (finetuned) models with 2B parameters in Table 1.
>
> > Q5: “There is no discussion of time or memory complexity of the methods”
>
> A5: This is not correct. Section 3.3 contains a paragraph “Computational Costs” (L244-249) which clearly outlines the complexity of all methods that are used. Also, L205-208 discusses the complexity of the proposed sequence-level methods. In short, token-level methods have the lowest computational cost for the MBR estimation but require evaluating |M|-many models in parallel. Sequence-level models have higher costs for MBR estimation but the hypothesis sets can be created independently. We have added an explicit paragraph header “Computational Costs” also to Section 3.2.
>
> > Q6: “I expect that these methods require significant computation.”
>
> A6: While MBR can be more costly than e.g. greedy decoding, our methods provide a principled way of trading test-time compute for better performance. This is in line with current trends, for example, the “thinking-mechanism” of GPT-o1 (https://openai.com/index/introducing-openai-o1-preview/). Furthermore, our methods can provide better performance than MBR with one model even given the same amount of computation.
>
> ## References
>
> [1] Vaswani, A. "Attention is all you need." Advances in Neural Information Processing Systems (2017).

---

> > ### Comment · Reviewer_Xat4 · 2024-11-20
> > **Thank you for the response**
> >
> > I have read the response, thank you for the clarifications.
> >
> > And apologies for missing the discussion of computational costs. To clarify my comment, I meant that I hoped for an empirical evaluation of the computational costs to give an idea of practically how large the hit is for the various methods.

---

> > > ### Author Response · Authors · 2024-11-21
> > > **Thank you for your response!**
> > >
> > > Dear Reviewer,
> > >
> > > Thank you for your response.
> > > We have added empirical measurements of decoding times to our response to Reviewer xUiD (which we summarize below) and will consider adding a further ablation to the paper.
> > >
> > > For running MBR on the full test set of E2E NLG for Table 1 we get the following times for running the MBR utility function evaluations:
> > >
> > > |  MBR@mean | Eq. 9  |  Eq. 10  |
> > > |---|---|---|
> > > |  402s | 408s  | 408s  |
> > >
> > > and for creating the hypothesis sets:
> > >
> > > |  MBR@mean | Eq. 9  |  Eq. 10  |
> > > |---|---|---|
> > > |  5824s | 5472s  | 8936s  |
> > >
> > > We use the unimodal posterior which gives the following results in terms of ROUGE-L:
> > >
> > > |  MBR@mean | Eq. 9  |  Eq. 10  |
> > > |---|---|---|
> > > |  45.16 | 45.57  |  45.38 |

---

> > > > ### Author Response · Authors · 2024-12-01
> > > > **Gentle Reminder.**
> > > >
> > > > Dear Reviewer, since the discussion period will be closing soon, please let us know if there are any further questions.

---

### Author Response · Authors · 2024-12-03
**Thank you for your reviews!**

We would like to thank the reviewers for their helpful comments. We feel encouraged that the reviewers agree that our work is well motivated, sound, and provides consistent improvements across extensive experiments.

The main concerns seem to be:

(1) Writing clarity.

(2) Computational burden.

We have addressed (1) by improving our manuscript and will continue to improve it towards a clearer presentation. We have addressed (2) by highlighting in our responses that MBR can be used to trade test time compute for better performance and is still useful for distillation, when decoding needs to be fast. In addition, we have provided further ablations on inference times. We will continue to take the reviewers’ suggestions into account to improve our work.

---

### Meta-Review · Area_Chair_K9nG · 2024-12-14

**Metareview:**

This work addresses undesirable outputs in language models by utilizing Minimum Bayes' Risk (MBR) decoding, which incorporates model uncertainty. By integrating a posterior distribution over model parameters into the expected risk calculation, the proposed approach effectively improves output selection and introduces a mechanism for abstention when appropriate.

The authors have addressed several concerns raised by reviewers, including clarifications on methodology and experimental details, and have significantly improved the manuscript. While some issues remain, the authors have proposed potential strategies to further strengthen their methods, demonstrating the value and potential impact of their work.

Therefore, I recommend accepting the manuscript, acknowledging its contributions and the authors’ commitment to refinement.

**Additional Comments On Reviewer Discussion:**

Most of the concerns revolve around efficiency and clarity.

Reviewer nT47 acknowledged that the authors have resolved some of his/her concerns. However, there are differing opinions on the suitability of MBR decoding for test-time computation. Nonetheless, I believe this is not a critical issue that would hinder the acceptance of this paper. Besides Reviewer TRc3 feels like this work has an interesting contribution in the field.

---

### Decision · Program_Chairs · 2025-01-22

Accept (Poster)